# Can We Ditch Feature Engineering? End-to-End Deep Learning for Affect Recognition from Physiological Sensor Data

**DOI:** 10.3390/s20226535

**Published:** 2020-11-16

**Authors:** Maciej Dzieżyc, Martin Gjoreski, Przemysław Kazienko, Stanisław Saganowski, Matjaž Gams

**Affiliations:** 1Department of Computational Intelligence, Wrocław University of Science and Technology, 50-370 Wrocław, Poland; kazienko@pwr.edu.pl (P.K.); stanislaw.saganowski@pwr.edu.pl (S.S.); 2Faculty of Computer Science and Management, Wrocław University of Science and Technology, 50-370 Wrocław, Poland; 3Jožef Stefan Institute, 1000 Ljubljana, Slovenia; martin.gjoreski@ijs.si (M.G.); matjaz.gams@ijs.si (M.G.); 4Jožef Stefan Postgraduate School, 1000 Ljubljana, Slovenia

**Keywords:** deep learning, multimodal deep learning, end-to-end machine learning, affect recognition, emotion recognition, stress detection, wearables, physiological signals, personal sensors

## Abstract

To further extend the applicability of wearable sensors in various domains such as mobile health systems and the automotive industry, new methods for accurately extracting subtle physiological information from these wearable sensors are required. However, the extraction of valuable information from physiological signals is still challenging—smartphones can count steps and compute heart rate, but they cannot recognize emotions and related affective states. This study analyzes the possibility of using end-to-end multimodal deep learning (DL) methods for affect recognition. Ten end-to-end DL architectures are compared on four different datasets with diverse raw physiological signals used for affect recognition, including emotional and stress states. The DL architectures specialized for time-series classification were enhanced to simultaneously facilitate learning from multiple sensors, each having their own sampling frequency. To enable fair comparison among the different DL architectures, Bayesian optimization was used for hyperparameter tuning. The experimental results showed that the performance of the models depends on the intensity of the physiological response induced by the affective stimuli, i.e., the DL models recognize stress induced by the Trier Social Stress Test more successfully than they recognize emotional changes induced by watching affective content, e.g., funny videos. Additionally, the results showed that the CNN-based architectures might be more suitable than LSTM-based architectures for affect recognition from physiological sensors.

## 1. Introduction

Emotions are complex states that result in psychological and physiological changes that influence our behaving and thinking [1]. The main assumption is that there are objectively measurable physiological responses to the autonomic nervous system activity that can be used for recognizing the human emotional state [2]. For example, the emotional state of fear usually initiates rapid heartbeat, rapid breathing, sweating, and muscle tension. These physiological changes can be captured by sensors embedded into wearable devices that can measure [3]:electrocardiography (ECG), which represents cardiac electrical activity,electroencephalography (EEG), brain electrical activity,electromyography (EMG), muscle activity,Blood Volume Pulse (BVP), cardiovascular dynamics,Electrodermal activity (EDA), sweating level,electrooculography (EOG), eye movements,respiration rate (RESP),facial muscle activation (EMO), emotional activation, andbody temperature (TEMP).

Signals were provided by some behavioral sensors such as an accelerometer (ACC), a gyroscope (GYRO), and environmental sensors. A barometer, an altimeter, ambient light, temperature sensors, and GPS may also be useful as additional data sources.

With the advancement of technology and the penetration of information systems into our everyday life, the emotional awareness of systems is becoming crucial. For example, in the domain of human–computer interaction (HCI), an emotion-aware system would enable a more natural interaction and better user experience [4]. In the mobile health domain, a system for monitoring affective states can contribute to the timely detection and treatment of emotional and mental disorders such as depression, bipolar disorders, and post-traumatic stress disorder (PTSD) [5]. Affect recognition systems can also be beneficial from an economic point of view. For example, the cost of work-related depression in Europe has been estimated to be €617 billion annually (EU report on mental health: https://ec.europa.eu/healt).

Unfortunately, three decades after establishing Affective Computing as a scientific field, emotionally intelligent systems are still not part of our everyday life. One reason is that relations between the wearable sensor data and the human psychophysiological states are not as explicit as is the relation between the wearable sensor data and human physical states. For example, smartphones can count steps and recognize human physical activities (e.g., running vs. walking) [6] but cannot recognize emotions and related affective states (e.g., cognitive load) with high accuracy [7,8].

In the last decade, deep learning (DL) dominated the artificial intelligence (AI) domain by achieving breakthroughs in image processing [9,10,11,12], natural language processing [13,14,15], and reinforcement learning [16,17]. Conversely to those domains, DL methods for sensor data are relatively scarce, and the appropriate DL methods and architectures are yet to be discovered [18]. In particular, the end-to-end approach, which can significantly simplify the classification flow, is not well investigated.

The classical feature-based approach requires domain-specific, expert knowledge about the sensors and signals to extract meaningful and informative features. For example, the R-peaks are used to compute the R-R Intervals (RRI) from the ECG signal, while the systolic peaks, maximum slopes, or onsets can be used to obtain the Heart Rate Variability (HRV) from the BVP signal [19]. Furthermore, feature extraction is time and computationally demanding, signal-dependent (each signal requires dedicated features), nonstandardized, and unsystematic (the number of features for each signal can vary from only a few for the skin temperature to hundreds for the EEG signal). To extract features, an advanced signal preprocessing is very often necessary. What is more, a great number of extracted features lead to the curse of dimensionality and require feature selection and feature reduction stages, which, in turn, may lead to the loss of information.

On the other hand, the end-to-end deep learning approach assumes that deep neural network architectures are able to extract the necessary information on their own, thus significantly reducing the complexity of the flow and amount of the work but also increasing the chance of extracting the appropriate information, unobtainable even for the domain experts. See Figure 1 for an illustrative comparison between the classical feature-based approach and the end-to-end DL techniques to the affect recognition problem.

Having that in mind, we present in this paper an extensive analysis of the end-to-end deep learning architectures for affect recognition from physiological sensor data. Our study has been inspired by the recent work by Ismail et al. [20], who presented a comprehensive overview of DL methods for time-series classification.

The main contribution of our work is as follows:Enhanced existing DL methods for time-series classification to work with multimodal data, i.e., the methods can simultaneously learn from several data sources (sensors), each having its own sampling frequency.Extensive comparison of ten end-to-end DL architectures for affect recognition on four publicly available datasets, three for emotion recognition, and one for stress recognition. To enable a fair comparison between the methods, Bayesian optimization was utilized.The implementation of the deep learning architectures (the source code), alongside the processed datasets, which are publicly available to enable replication, comparison, and further research.

## 2. Related Work

In the next subsections, we present an overview of studies on emotion and stress recognition. The focus is on research using physiological data from wearable sensors.

### 2.1. Feature-Based Machine Learning

Most of the studies so far have applied various feature-based machine learning (ML) approaches, i.e., the valuable features are extracted from signals and processed based on expert knowledge to train classifiers for affect recognition (Figure 1). Iacoviello et al. [21] combined discrete wavelet transformation, principal component analysis, and support vector machine (SVM) to build a hybrid classification framework using EEG signals. In turn, Khezri et al. [22] combined EEG with GSR to recognize six basic emotions via k-nearest neighbors (kNN) classifiers. Mehmood and Lee [23] utilized independent component analysis to extract emotional indicators from EEG, EMG, EDA, ECG, and the effective refractory period (ERP). Mikuckas et al. [24] presented an HCI system for emotion recognition that uses spectrotemporal analysis based solely on R-R signals. More specifically, they focused on recognizing stressful states utilizing heart rate variability (HRV) processing. Yin et al. [25] developed an approach for emotion recognition from physiological signals using an ensemble of autoencoders. They used features from seven physiological sensors: EEG, EDA, EOG, EMG, ECG, BVP, and the respiration rate sensor as an input. In general, the methods based on EEG data usually outperform the ones based on other data [26,27]. It probably comes from the fact that EEG provides a more direct channel to one’s mind.

The pioneers in the field of stress detection using physiological sensors and ML are Healey and Picard [28], who proposed a quite accurate stress detection system in 2005. It achieved an accuracy as high as 97% when tested in a constrained real-life scenario, i.e., subjects driving a car. They used features extracted from ECG, EMG, EDA, and respiration to feed the input of the system. Even though the presented system was obtrusive, it confirmed that stress detection was possible in a real-life scenario even in 2005. Hovsepian et al. [29] proposed cStress, a method for continuous stress assessment in real life using chest-worn ECG sensors. Later on, cStress was used in another real-life study [30]. Gjoreski et al. proposed a context-based method for real-life stress monitoring using wrist devices [31].

Nakisa at al. [32] collected EEG and BVP signals in a study involving 20 participants who watched nine video clips—emotional stimuli. SVM, Multilayer Perceptron, and Long-Short Term Memory (LSTM) were tested for emotion recognition. They suggested a new framework for hyperparameters optimization—Differential Evolution (DE). DE performed better after the same number of iterations than other approaches (gain by 0-8 pp)—Particle Swarm Optimization (PSO), Simulated Annealing (SA), Tree-structured Parzen Estimator Approach (TPE), and Random Search; however, DE’s execution time was about twice as long.

All of these systems for emotion and stress recognition are based on features extracted from signal data provided by the wearable sensors. The advanced domain-dependent signal processing and transformation (e.g., wavelet/Fourier transform, heart rate extraction from BVP, and spectral analysis), various feature extraction (including signal morphology and statistical and nonlinear measures), as well as feature selection and reduction are the most important and most challenging steps in the overall processing pipeline (Figure 1, the upper flow). As an opposite idea, the end-to-end deep learning offers the possibility to avoid feature extraction and most signal/feature processing steps by learning directly from the sensor signal data.

### 2.2. End-to-End Learning

DL represents a class of ML algorithms that use a cascade of multiple layers of nonlinear processing units, which are typically neurons [33]. The first layer receives the input data, and each successive layer accepts the output from the previous layer as its input. A common advantage of the DL models based on Convolutional Neural Networks (CNNs) and LSTMs is that they can learn directly from the raw data, thus avoiding the need for signal and feature processing (see Figure 1, the lower flow). This characteristic makes end-to-end DL particularly interesting since it holds the potential to discover useful patterns in the original physiological data.

Martinez at al. were probably the first to introduce end-to-end DL for affect recognition from physiological signals [34]. They used a combination of denoising autoencoders, CNNs, preference learning, and automatic feature selection on The Maze-ball dataset. This dataset consists of BVP and EDA data collected from game players, together with questionnaires. The authors also used ad-hoc feature extraction as the baseline. DNNs trained on EDA signals were better at recognizing affective states (anxious, exciting, frustrating, fun, and relaxing) than the baseline for almost all affective states. For BVP, the results were mixed and did not show the superiority of deep learned features over statistical features. When signals where combined, DNNs achieved above baseline accuracy for all affective states.

Keren at al. [35] claim to conduct the first study on end-to-end DL for emotion recognition based on physiological signals (ECG and EDA) using neural networks (NNs). The authors measured the performance of the models using the concordance correlation coefficient. Hyperparameter optimization was performed by hand—no specific method was reported. The method predicted valence and arousal as a classification task with categories ranging from −0.30 to 0.30 with a 0.01 step. A concordance correlation coefficient of 0.430 for arousal and of 0.407 for valance was reported on a separate test set.

End-to-end DL was utilized in the flow state recognition task, which was defined as ”affective state of optimal experience, total immersion and high productivity” [36]. Based on BVP and EDA signals from Emaptica E4, a model was trained to recognize two (high/low flow level) or three flow states (boredom, stress, and flow). An accuracy of 0.68 was achieved for the 2-class problem (based on BVP only) and of 0.49 for the 3-class problem (BVP and EDA combined).

Schmidt et al. [37] applied end-to-end learning to the dataset gathered in the wild. Eleven subjects wore Empatica E4 for 16 days and filled out the self-assessment questionnaires using Ecological Momentary Assessments (EMA), i.e., every 2 hours ±30 mins. They used feature extraction for PPG, ACC, EDA, and TEMP signals described in [38] for four classification tasks: three levels of arousal, valence, and anxiety, as well as binary stress recognition. They tested several CNN end-to-end solutions, including an autoencoder and multitask classification. The end-to-end F1 results were, on average, 1.8% better than classical feature-based approaches, showing the potential of the end-to-end approach. Unfortunately, their dataset and code are not publicly available.

In a study of the valance level during walking in the city center, participants filled Self-Assessment Mankin (SAM) questionnaires [5]. The collected data, including heart rate, EDA, body temperature, and motion data, was put into three different end-to-end DL architectures. Multi-Layer Perceptron achieved an F1-score of 0.63, the CNNs achieved 0.71, and the CNN-LSTM achieved 0.874.

Recently, a Bayesian DL framework for high/low valance recognition from inter-beat-intervals (IBI) was proposed in [39]. The DL architecture was comprised of CNNs and bidirectional LSTMs and was tested on two datasets: AMIGOS and DREAMER. The proposed approach yielded F1-scores of 0.88 for AMIGOS and 0.83 for DREAMER. However, some samples were not classified because of the restriction on the confidence level for predictions. For the non-Bayesian approach, the F1-scores were 0.78 and 0.68, respectively (all samples classified). This architecture was also applied to IBI signals from Garmin Vivosmart 3, achieving a peak F1-score of 0.7 [40], but the classification was performed only for about 40% samples. For all samples, the F1-score only exceeded 0.6.

Another approach to end-to-end DL was presented by Li at al. [41]. Firstly, they transformed each raw signal into a spectrogram. Later, these spectrograms were fed into an attention-based bidirectional LSTM network and then to an unspecified DNN. Their network achieved an F1-score of 0.72 for binary arousal recognition and 0.70 for binary valence evaluation.

Qiu at al. [42] proposed Correleted Attention Networks for emotion recognition using bidirectional Gated Recurrent Units (GRUs), a Canonical Correlation Layer, a Signal Fusion Layer, an Attention Layer, and a Classification Layer. This architecture was tested on three datasets: SEED, SEED IV, and DEAP. Their framework achieved higher accuracy than feature-based SVM, although details about the features utilized were not provided.

Additionally, CNNs were used on the MAHNOB-HCI dataset to achieve better accuracies than those found using the methods based on feature extraction [43].

### 2.3. Summary

The main drawbacks of the aforementioned end-to-end DL methods for affect recognition are as follows:Their raw data and/or source code have not been published.Only one deep learning architecture/approach (or method variations of the authors) is considered (an exception is [5], analyzing three architectures, and [37], considering multitask vs. single-task models along with two autoencoder versions).The results, if compared, refer to old feature-based approaches.The outputs (affective state classes) are incomparable between studies, usually solving a binary and sometimes multiclass problem.Hyperparameters are hand-picked, not systematically tuned.

All of these drawbacks make research replicability infeasible. Moreover, there has been no study comparing various end-to-end architectures on multiple and multimodal affective datasets. We seek to fill this gap in this paper.

## 3. Materials and Methods

DL architectures for signal processing have not yet realized any outstanding breakthrough, and designing them remains challenging, especially for problems with limited data for training. The main aim of the experiments was to compare ten end-to-end DL architectures (see Section 3.1):Fully convolutional network (FCN) [44],Residual network (Resnet) [45],Multi layer perceptron (MLP) [44],Encoder [46],Time convolutional neural network (Time-CNN) [47],Multichannel deep convolutional neural network (MCDCNN) [47],Spectrotemporal residual network (Stresnet) [48],Convolutional neural network with long-short term memory (CNN-LSTM) [5],Multi layer perceptron with long-short term memory (MLP-LSTM), andInceptionTime [49].

Architectures 1–6 were taken from a review of DL architectures for time-series classification [20]. We enhanced them for multimodal data. Twiesen architecture was not implemented in Keras, so it was omitted. t-LeNet and MCNN were the worst performing models, so they were left out. In our comparison, we also included the Stresnet architecture from our prior work, as well as InceptionTime [49]. Additionally, we implemented two LSTM networks, one of which (CNN-LSTM) was inspired by [5].

All the above architectures were tested on four reference emotion and affect datasets (Section 3.4) as an emotion/affect classification task (Section 3.6). To enable fair comparison, hyperparameters of each architecture were tuned (Section 3.3), and each architecture was validated using 5-fold subject-independent cross-validation (Section 3.7). The details of implementation together with a publicly available code are described in Section 3.8.

### 3.1. Deep Learning Architectures

The layer-based structure of the NNs facilitates the construction of a variety of DL architectures by combining layers. For example, CNN layers can be stacked on top of LSTM layers, namely, the input is received by the CNN layers and propagated to the LSTM layers. In addition to the vertical stacking, one can also experiment with horizontal branching. For example, for a 2-signal dataset (2 modes), one can use a separate DL branch for each signal (mode) and later fuse the two outputs. Such fusion can simply be performed by using the concatenation layer and its further processing with other layers, e.g., a fully connected (dense) layer.

Which DL architecture is most suitable for end-to-end learning on multimodal physiological signals may depend on the dataset; thus, extensive experimentation is required [48]. They are described in the subsequent sections.

The three best-performing architectures are presented in Figure 2: the spectrotemporal residual network (Stresnet), the fully convolutional neural network (FCN), and the residual network (Resnet). Additionally, Table 1 contains a short summary of all DL architectures used in the experiments.

MLP contains *d* fully connected layers (FCL; dense) for each signal/mode (*N* signals) and a final FCL, which provides the output. FCN consists of three convolutional layers (CLs) for each signal and the final FCL layer. The Encoder is similar to FCN but with an additional attention layer (Att) between the final CLs and the FCL. Time-CNN is quite similar to FCN architecture. CNN-LSTM was partially based on the description of architecture presented in [5]. MLP-LSTM extends the MLP with an additional LSTM layer. MCDCNN trains in two parallel CNN layers for each modality (sensor signal). The last three architectures, Resnet, Stresnet, and Inception are the most complex ones. They all are based on residual connections, i.e., short-cut connections that bind two nonconsecutive layers and reduce the vanishing gradient problem [50]. The Resnet contains *N* branches (one branch per signal) that have *d* residual blocks (e.g., three residual blocks), and each block contains 3 CLs. Stresnet is a network in which each signal is associated with two branches: a Resnet that analyzes the raw sensor signal in the time domain and another branch processing a spectral representation of the signal. Towards the end of the network, the two branches of each signal, namely, the spectral and the temporal ones, are merged using FCL. The Inception network, similarly to Resnet and Stresnet, includes the residual blocks, but it additionally uses Inception modules that (1) apply multiple filters of different sizes simultaneously to the same input; and (2) exploit “bottleneck” layers, which reduce the dimensionality of the input as well as the model’s complexity and thus potentially avoid overfitting problems for small datasets.

### 3.2. Bayesian Optimization Methods

When the term hyperparameters is used, it usually refers to parameters of the model (e.g., DL architecture), which is trained in contrast to parameters (weights) derived directly from the training dataset.

Very often, hyperparameters are chosen by a human supervisor based on their expert knowledge and experience. Alternatively, a random search is often performed. It evaluates a randomly chosen configuration *x* from a set of all correct configurations X. Moreover, a grid search can be applied. It tests all possible configurations from X in a given sequence.

In contrast, Bayesian optimization is a process of evaluating possible configurations, but each following configuration is picked based on the history H of the previous evaluations.

Bayes’ theorem describes a way of calculating the conditional probability of event A happening assuming that B is true, i.e., P(A∣B). It assumes that knowledge about probability of observing events A (P(A)) and B (P(B)) while also being able to assess the probability of event B assuming that A is true: P(B∣A). It is also sometimes referred to as a measure of the “degree of belief” in a given statement [51]. In many cases, the component probabilities are easy to obtain. Mathematically, it is stated as [51]:(1)$$P\left(A|B\right)=\frac{P(B|A)P(A)}{P(B)}.$$

Sequential Model-based Global Optimization (SMBO) algorithms optimize configuration x∈X for a given model for which we can compute a fitness function (f:X→R). This class of algorithms picks a “promising” configuration $x^{*}$ in each iteration based on history H of the already evaluated configurations with their respective results of calculated fitness function, i.e., $H=({\left(x_{i}^{*},f\left(x_{i}^{*}\right)\right)}_{i=0}^{n}$ [52]. Therefore, they can be seen as an implementation of the Bayesian optimization.

Tree-structured Parzen Estimator approach (TPE) is an SMBO algorithm, which returns a set of configurations with the highest Expected Improvement beyond a given threshold $y^{*}$. Instead of directly calculating the probability of a given score *y* for a given configuration *x* (P(x∣y)), it computes P(x), P(y), and P(y∣x) [52].

In this study, the TPE method was utilized for hyperparameter optimization. This kind of Bayesian optimization enables us a more fair comparison of the methods because

it exploits history to make more informed guesses about hyperparameters,it does not need or use any prior knowledge about tuned architectures, andthe hyperparameter search space is defined at the beginning of the experiments, which prevents the researchers from making a tuning decision based on their preliminary test results, thus reducing the researchers’ bias and minimizing the possibility of overfitting [53,54].

### 3.3. Hyperparameters Search Space

The hyperparameters search space ensures that the default hyperparameters (i.e., the hyperparameters used in previous studies) are in the center of the search space for each architecture. These default hyperparameters were taken from said studies [5,20,48]. Please note that the hyperparameter set was fixed and actually determines the architecture itself.

Each parameter was given an interval of values, which, based on our previous experience, had a chance to significantly influence architecture evaluation scores.

In order to describe the hyperparameter search space, the null distribution specification language [55] was used with two extensions:{A,B}—the joint distribution of two null distributions related to expressions *A* and *B*;${\left\{A_{i}\right\}}_{i=1}^{n}=\left\{A_{1},...,A_{n}\right\}$—the joint distribution of *n* null distributions referring to *n* signals, e.g., ECG, BVP, EEG, EDA, RESP, TEMP, and ACC.

Hyperparameters defined by the second notation enable us to adjust the hyperparameters separately for each signal. Therefore, the inferred hyperparameters can be tailored according to the profile and importance of each signal.

The individual hyperparameters are as follows:$dense\_output_{i}$: a dimensionality of the output space of dense layers for signal *i*$filters_{i}$: a dimensionality of the output space of convolution layers for signal *i*$filters\_multiplier_{i}$: a multiplier of a dimensionality of the output space of convolution layers for signal *i*, applicable when different filters are used in different layers.$kernel\_size\_multiplier_{i}$: a multiplier specifying the length of the convolution window for signal *i*, used when different layers have different kernel sizes.$lstm\_units_{i}$: output space of the LSTM layer for signal *i*.depth: for Resnet and Stresnet, the number of residual blocks; for Inception, the number of inception blocks.

For MLP and MLP-LSTM, the following hyperparameter search space was considered (n+1 hyperparameters):(2)$$\begin{array}{c} A=\{{\left\{dense\_output_{i}=\mathrm{choice}\left(250,500,1000\right)\right\}}_{i=1}^{n},\\ depth=\mathrm{choice}(3,4,5)\} \end{array}$$
where choice(3,4,5) means that the hyperparameter depth may be chosen from three possible values: 3,4,5.

For MCDCNN, Time-CNN, Encoder, and FCN, there are 2n hyperparameters:(3)$$\begin{array}{c} B=\{{\left\{filters\_multiplier_{i}=\mathrm{choice}\left(0.5,1,2\right)\right\}}_{i=1}^{n},\\ {\left\{kernel\_size\_multiplier_{i}=\mathrm{choice}\left(0.5,1,2\right)\right\}}_{i=1}^{n}\} \end{array}$$

For CNN-LSTM (2n+n=3n hyperparameters),
(4)$$\begin{array}{c} C=\{B,{\left\{lstm\_units_{i}=\mathrm{choice}\left(0.5,1,2\right)\right\}}_{i=1}^{n}\} \end{array}$$

For Resnet (2n+1),
(5)$$\begin{array}{c} D=\{{\left\{filters_{i}=\mathrm{choice}\left(32,64,128\right)\right\}}_{i=1}^{n},\\ {\left\{kernel\_size\_multiplier_{i}=\mathrm{choice}\left(0.5,1,2\right)\right\}}_{i=1}^{n},\\ depth=\mathrm{choice}(2,3,4)\} \end{array}$$

For Inception (2n+1),
(6)$$\begin{array}{c} E=\{{\left\{filters_{i}=\mathrm{choice}\left(16,32,64\right)\right\}}_{i=1}^{n},\\ {\left\{kernel\_size_{i}=\mathrm{choice}\left(21,41,81\right)\right\}}_{i=1}^{n},\\ depth=\mathrm{choice}(5,6,7)\} \end{array}$$

For Stresnet (2n+1),
(7)$$\begin{array}{c} F=\{{\left\{filters_{i}=\mathrm{choice}\left(32,64,128\right)\right\}}_{i=1}^{n},\\ {\left\{kernel\_size\_multiplier_{i}=\mathrm{choice}\left(0.5,1,2\right)\right\}}_{i=1}^{n},\\ depth=\mathrm{choice}(5,6,7)\} \end{array}$$

Additionally, four hyperparameters related to the optimizer itself were tuned for each architecture:(8)$$\begin{array}{c} O=\{lr=\mathrm{randint}(-7,-1),\\ decay=\mathrm{choice}(0.001,0.0001,0.00001,0),\\ reduce\_lr\_factor=\mathrm{choice}(0.5,0.2,0.1),\\ reduce\_lr\_patience=\mathrm{choice}(5,10)\} \end{array}$$
where randint(−7,−1) means that hypeparameter lr can be an integer in the range [−7, −1); lr and decay are the learning rate ($10^{lr}$) and decay in the Adam optimizer, respectively; reduce_lr_factor is a factor by which the learning rate is reduced after reduce_lr_patience number of epochs with no improvement.

### 3.4. Datasets

Four datasets were included in our experiments. The first three, AMIGOS [56], ASCERTAIN [26], and DECAF [57], are related to emotion recognition. The fourth dataset, WESAD [38], focuses on stress recognition.

AMIGOS is a multimodal dataset for affect recognition, personality traits, and mood on individuals and groups. It contains data of 40 participants who watched 16 short affective videos (51–150 s each) and 37 participants who watched four long affective videos (14–24 min each). The participants’ signals, namely, EEG, ECG, and EDA, were recorded using wearable sensors, i.e., Emotiv EPOC Neuroheadset (EEG), Shimmer 2R (ECG), and an extension to the Shimmer 2R platform placed on the left hand’s middle and index fingers (EDA). Participants’ emotions were annotated with both self-assessment (valence, arousal, control, familiarity, liking, and basic emotions) as well as an external assessment of valence and arousal.

ASCERTAIN is a multimodal dataset for personality and affect recognition using commercial physiological sensors. It contains data from 58 subjects who watched 36 short videos (51–128 s each). The participants’ physiological signals were also recorded, including frontal EEG, ECG, EDA, and Facial Emotional Activation Features (EMO). Additionally, the participants rated each video in terms of the levels of arousal, valence, engagement, liking, and familiarity. The authors of the paper did not provide any information about which devices they used, except for description of their placement.

DECAF is a multimodal dataset for affect recognition. It contains data of 30 participants that watched 40 music-video segments (60 s each) and 36 movie clips (51–128 s each). It also contains EEG data, Facial Emotional Activation Features (EMO), horizontal Electrooculogram (EOG), ECG, and trapezius EMG. Additionally, the participants rated the affective stimuli in terms of the levels of arousal, valence, and dominance. ELEKTA Neuromag was used to record EEG, but other devices were not explicitly specified, except for their position on the body.

The WESAD dataset [38] was collected in a study focused on stress, where the subjects experienced both an emotional and stress stimuli. More specifically, WESAD contains data from 15 subjects. Each subject underwent three sessions: a baseline session (neutral reading task; 20 min), an amusement session (watching a set of funny video clips; 392 s), and a stress session (being exposed to the Trier Social Stress Test [58]; about 10 min). The amusement and stress sessions were followed by a guided meditation. The participants’ physiological response was recorded using both a wrist and chest device. The sensor data includes BVP, ECG, EDA, EMG, respiration, body temperature, and three-axis acceleration collected with two wearables, i.e., Empatica E4 (wristband) and RespiBAN (chest device).

### 3.5. Datasets Preprocessing

The signals synchronized by the datasets’ authors were exploited as an initial input to the methods we tested. Additionally, the sensor data from each dataset, each subject, and each signal was preprocessed using the following:3–97% winsorization, which removes extreme values form the signal data;Butterworth low-pass filter with a 10 Hz cut-off which removes components above the threshold frequency of 10 Hz;downsampling, which reduces the dimensionality of the inputs and consequently decreases the number of learning parameters in the DL models (see Table 2 for more details);min-max normalization.

For the emotion datasets (AMIGOS, ASCERTAIN, and DECAF), each subject had many short sessions, in which the affective stimuli were presented. Each session was used as an input window to the DL methods. Windows were created from the last 50 s of the signals from each session, as it was the minimum session length. The long sessions from AMIGOS were discarded.

In the WESAD dataset, each subject had only a few long sessions. To segment the long sessions into input windows, each signal was divided into 60 s windows with 30 s slides, i.e., each window overlapped the previous and next window for 30 s. Each window was assigned to the dominant class in its time span.

### 3.6. Classification Task

The ASCRTAIN, AMIGOS, and DECAF datasets contain information about self-reported valance and arousal. These values were mapped to 4 classes:*low-arousal-low-valence* (LALV),*low-arousal-high-valence* (LAHV),*high-arousal-low-valence* (HALV), and*high-arousal-high-valence* (HAHV).

This is a common method of discretization of Russell’s *arousal-valence* model [8]. Table 3 describes which values of arousal/valence where considered as high/low. The representation (imbalance) of classes is depicted in Table 4.

### 3.7. Validation

A 5-fold cross-validation was utilized. All datasets were pseudo-randomly divided into five test sets with approximately equal sizes. For each test set, the train and validation sets were created from the rest of the subjects. These subjects were pseudo-randomly split into a train and validation set with a ratio of 4:1. This assignment was constant during the whole experiment in order to discard any differences due to the better or worse division of the data.

The general goal of learning is to maximize performance on the training data. However, we utilized a validation set, separate from the test set, to avoid overfitting on the training data. The validation data was used for evaluation purposes of the trained model. The best performing models on the validation data were the final models applied to the test sets and are reported in Section 4.

For each dataset, architecture, test set, and given hyperparameter combination, five training iterations were conducted, and the results were averaged over them. During training, a maximum of 150 epochs were run with early stopping implemented with a patience of 15, i.e., if validation loss had not improved for 15 epochs, the training stopped.

### 3.8. Method Implementation

Methods were implemented in Python using Tensorflow 1.13.1, keras 2.2.4, SciPy 1.4.1, and scikit-learn 0.22.1.

Implementation of FCN, Resnet, Encoder, MCDCNN, Time-CNN, and MLP was based on code provided at https://github.com/hfawaz/dl-4-tsc. The rest of the architectures were omitted, as they were not implemented in Keras or performed poorly in the study [20].Implementation of Inception was based on code provided at https://github.com/hfawaz/InceptionTime.Implementation of Stresnet code was used from the previous work by Gjoreski at al. [48].Implementation of MLP-LSTM was based on the above MLP implementation.Implementation of CNN-LSTM was based on the description provided by Kanjo at al. [5].

Each architecture implementation (except for MCDCNN and Stresnet) was adopted for multimodal data by creating separate branches for each signal. Details are presented in Section 3.1. The code was made publicly available at https://github.com/Emognition/dl-4-tsc.

## 4. Results

To compare the results of different architectures, the macro F1-score was utilized. The F1-score (Equation (Equation 10)) is a harmonic mean of precision (Equation (Equation 11)) and recall (see Equation (Equation 12)). The macro F1-score is defined as the arithmetic mean for F1-scores of N classes calculated separately (Equation (Equation 9)). While reporting the Area Under the Receiver Operating Characteristic Curve (ROC AUC), we also used the ROC AUC averaged over all classes. There were five iterations of each fold from which we obtained the average F1-score, accuracy, ROC AUC, as well as standard deviations of these metrics. Then, all metrics from five folds were averaged and the final values are reported in Table 5, Table 6, Table 7 and Table 8.
(9)$$F1\;score\;macro=\frac{F1\;score_{1}+\ldots +F1\;score_{N}}{N}$$
(10)$$F1\;score=2\times \frac{Precision\times Recall}{Precision+Recall}$$
(11)$$Precision=\frac{True\;Positive}{True\;Positive+False\;Positive}$$
(12)$$Recall=\frac{True\;Positive}{True\;Positive+False\;Negative}$$

Table 5 presents the experimental results for the AMIGOS dataset. It can be seen that none of the DL classifiers achieved better F1-score results compared to the random guess classifiers. To the best of our knowledge, there is no study on the AMIGOS dataset that has incorporated the same 4-class problem. Harper at al. [39], using CNN with LSTM and heart rate, achieved an F1-score of 0.78 and an accuracy of 0.79 for high/low valence detection (but with different thresholds) for their non-Bayesian approach, i.e., for all samples. The authors of the dataset achieved F1-scores of 0.57 and 0.59 for valence and arousal, respectively, using only short films.

Table 6 presents the experimental results for the DECAF dataset. The best performing models are built using FCN, Stresnet and Encoder. These three models are slightly better than the random guess classifier. The highest F1-score of 0.26 was achieved by the FCN classifier. To the best of our knowledge, there is no study on the DECAF dataset with the same 4-class problem. The authors of the dataset reported F1-scores of 0.58 and 0.56 for arousal and valence independently. They applied classical feature extraction methods.

Table 7 presents the experimental results for the ASCERTAIN dataset. The three best performing models are built using Inception, Resnet, and FCN. We have not found any study on the ASCERTAIN dataset that addresses the same 4-class problem. In the original paper of the ASCERTAIN dataset [26], F1-scores were 0.71 and 0.67 for valence and arousal, respectively. They utilized classical feature extraction methods.

Table 8 presents the experimental results for the WESAD dataset. It can be seen that all of the DL classifiers outperformed the baseline classifiers (random guess and majority class). The best performing model achieved F1-score equal to 0.73 and is built using the FCN architecture. The authors of the dataset achieved an accuracy of 0.80 and an F1-score of 0.69 for all modalities using feature extraction, AdaBoost, and leave-one-subject-out validation [38]. Lin at al. achieved an accuracy of 0.83 using DL, but they used different input data (1 s windows) and a different validation method [59].

In Table 9, the detailed results for the best architecture for each dataset in our study are shown. There are substantial differences in F1-score between classes for some datasets.

Our results (Table 10) are consistent with the results originally obtained on multivariate time series by [20]. The only differences are between the rank of MLP and MCDCNN. In our study, the MCDCNN performed better than MLP.

## 5. Discussion

A general observation that can be made from the results for the emotion datasets (AMIGOS, ASCERTAIN, and DECAF), is that end-to-end DL classifiers performed rather poorly. This does not refer to stress recognition (the WESAD dataset), where the FCN DL classifier achieved an F1-score equal to 0.73 and an ROC AUC of 0.91. One reason for such behavior may be the fact that physiological responses induced by the emotional videos used in the emotion datasets are generally less significant than responses invoked by the Trier Social Stress Test used in WESAD. The difficulty of recognizing subtle affective stimuli using physiological sensors has also been recognized in the related work [7]. This is also confirmed by the baseline results reported by the creators of the datasets, i.e., the F1-score for binary classification of low vs. high arousal and low vs. high valence are between 0.55 and 0.60 [26,56,57]. The relatively low separation of binary classes is also visible in Figure 4 in [56], where classes are very close to each other. On the contrary, the WESAD dataset contains the class-label *stress*, which is easier to be recognized compared to the class labels defined only by the arousal and valence levels in the emotion datasets.

Therefore, our classifier built using the multimodal end-to-end FCN architecture achieved results comparable to the original WESAD paper. It reveals that end-to-end DL can be efficiently used instead of the classical feature-based approaches for some affect recognition tasks operating on physiological data. It is also in line with the results from [37].

It is also worth noting that, among the “simple” CNNs (i.e., FCN, Time-CNN, and MCDCNN), the best performing was FCN, which includes layers with the highest number of filters. It may indicate that DL architectures potentially need to accommodate many features to fully learn a signal representation. Furthermore, consistency with the results presented in [20] may indicate that there might be more to incorporate from time series classification into affective computing and end-to-end DL physiological signals classification in order to achieve better results.

Regarding different DL architectures tested by us, CNN layers outperformed the architectures based on LSTM layers in all of the experiments. This indicates that CNN-based architectures may be more suitable than LSTM for affect recognition from physiological signals. This is counterintuitive, especially because the LSTMs were originally developed for handling sequences in the input data. Our intuition for such results is that the CNNs may be more robust to noise, e.g., the CNN filters can perform a moving average on the input sensor data; thus, they can remove some noise. The above raises a new question: can additional preprocessing methods increase DL performance? It, however, goes in the opposite direction than the general idea of end-to-end learning: put anything in the input and let the model deal with that.

In our work, we carried out simple preprocessing tasks such as winsorization, downsampling, filtering, and normalization (see Section 3.5). We are aware that the application of the low-pass filter with a 10 Hz threshold could have removed some potentially valuable information, especially for EEG signals, e.g., beta waves belong to the 15–32 Hz range and gamma ≥ 32 Hz. On the other hand, we wanted to test DL architectures in the signal-agnostic environment, i.e., without any specific knowledge about individual signals. This enabled us to test end-to-end multimodal approaches, in which each signal was processed by a separate deep neural network branch (Figure 2).

In principle, there is an open issue as to what *end-to-end machine learning* is and which processing tasks, if any, may be performed before model training. Yet another problem for deep learning models are the quantities of the learning samples. In total, we had at our disposal only 620–2280 cases, depending on the dataset (Table 4). It is a rather insufficient amount of data for properly training deep learning architectures.

Concluding, the main benefit of end-to-end learning is that expert knowledge about the nature of the physiological signals and their relations to affective states are not necessary (Figure 1). Signals may be processed by the same types of branches, which, while learning, adapt to the specific profile of individual signals.

This work establishes a base for further development of end-to-end DL architectures for affect recognition from physiological sensor data. Building on this work, one can easily analyze the influence of different signal (pre)processing techniques or the impact of signal selection. For example, Gjoreski et al. showed that Stresnet builds better models for monitoring driving distractions and monitoring human locomotion modes when the input sensor modalities are preselected, e.g., using information gain [6,48].

## 6. Conclusions

Ten end-to-end DL architectures on four different datasets for affect recognition were compared in this study. The source data included multimodal physiological signals for emotion and stress recognition. The existing DL architectures specialized for time-series classification were enhanced to enable learning from several sensors simultaneously in an end-to-end manner. Each input sensor data (mode) was utilized with its sampling frequency, keeping all information hidden in the signals without any advanced preprocessing or feature extraction. Additionally, to enable fair comparison among the different DL architectures, Bayesian optimization was used for hyperparameter tuning.

The experimental results showed that the performance of the models depends on the intensity of the physiological response induced by the affective stimuli. For example, the DL models recognize stress induced by the Trier Social Stress Test more successfully than they recognize emotional changes induced by watching affective content, e.g., funny videos. Additionally, the results revealed that the CNN-based architectures may be more suitable than LSTM-based ones for affect recognition from physiological sensors. Finally, the enhanced multimodal FCN architecture achieved comparable results for stress recognition using only raw signals, omitting the need for feature extraction.

## Figures and Tables

**Figure 1 sensors-20-06535-f001:**
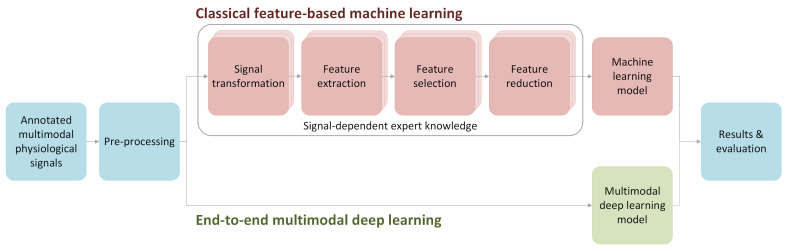
The comparison between the classical feature-based approach to multimodal physiological signal processing (red) and the end-to-end deep learning workflow (green), both applicable to affect recognition problem.

**Figure 2 sensors-20-06535-f002:**
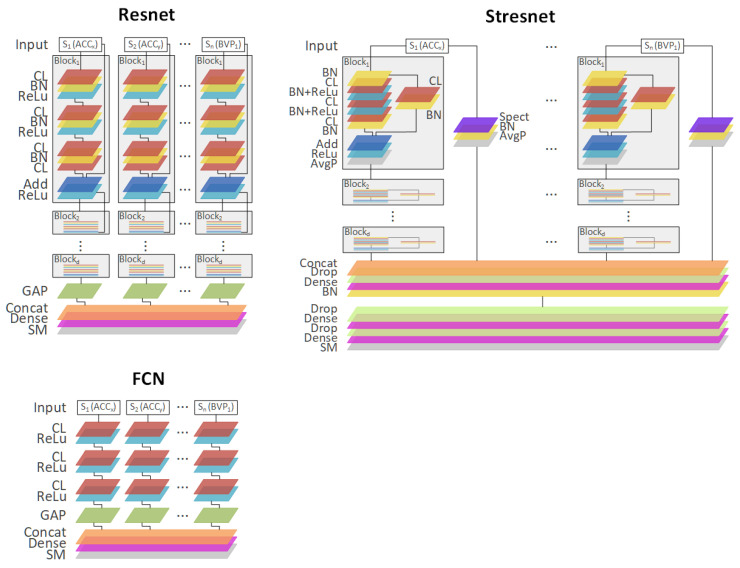
Three best performing end-to-end DL architectures: spectrotemporal residual networks (Stresnet), a fully convolutional neural network FCN), and a residual network Resnet). All architectures consist of layers stacked vertically as well as horizontal branches separately dedicated for each signal. The branches are finally concatenated (Concat) and fed into a dense layer (Dense) with softmax activation (SM).CL: convolutional layer, ReLu: rectifier layer, BN: batch normalization layer, GAP: global average pooling layer, Add: addition layer, Drop: dropout layer, AvgP: average pooling layer, Spect: spectogram layer.

**Table 1 sensors-20-06535-t001:** Summary of the deep learning architectures used in the experiments with respect to different DL layers (N: number of signals, FC: fully connected (dense) layer, CL: convolutional layer, LSTM: Long Short-term Memory, ResBloc: residual block with three CLs, Att: attention mechanism). FCN, Resnet, and Stresnet are shown in Figure 2 in greater depth.

Architecture	Description
FCN [44]	N x [CL - CL - CL] - FC
Resnet [45]	N x [ResBlock - …- ResBlock] - FC
Stresnet [48]	N x [ResBlock-Time + ResBlock-Freq] - FC
MLP [44]	N x [FC - …- FC] - FC
Encoder [46]	N x [CL - CL - CL - Att] - FC
Time-CNN [47]	N x [CL - CL] - FC
CNN-LSTM	N x [CL - CL - LSTM] - FC
MLP-LSTM	N x [FC - …- FC - LSTM] - FC
MCDCNN [47]	N x [CL - CL] - FC - FC
Inception [49]	N x [*Inception module*] - FC

**Table 2 sensors-20-06535-t002:** Sampling frequencies for each modality before (original sampling) and after downsampling. The total number of distinct signals is provided for each dataset. For example, ACC modality collected from Empatica and RespiBAN in the WESAD dataset consists of three signals: $ACC_{x},ACC_{y},andACC_{z}$. Therefore, WESAD contains eight one-signal modalities and two modalities with three signals each.

Dataset	Modes of Signals (Distinct Signals/DL Branches)	Original Sampling	Downsampled to
WESAD (14 signals)	ECG RespiBAN	700 Hz	70 Hz
	ACC RespiBAN (3 signals)	700 Hz	10 Hz
	EMG RespiBAN	700 Hz	10 Hz
	EDA RespiBAN	700 Hz	3.5 Hz
	TEMP RespiBAN	700 Hz	3.5 Hz
	Respiration RespiBAN	700 Hz	3.5 Hz
	BVP Empatica	64 Hz	64 Hz
	ACC Empatica (3)	32 Hz	8 Hz
	EDA Empatica	4 Hz	4 Hz
	TEMP Empatica	4 Hz	4 Hz
AMIGOS (17)	ECG (2)	128 Hz	64 Hz
	EEG (14)	128 Hz	64 Hz
	EDA	128 Hz	8 Hz
ASCERTAIN (33)	ECG (2)	128 Hz	64 Hz
	EEG (8)	32 Hz	32 Hz
	EDA	128 Hz	8 Hz
	EMO (22)	128 Hz	4 Hz
DECAF(25)	ECG	1000 Hz	64 Hz
	EMG	1000 Hz	64 Hz
	EOG	1000 Hz	64 Hz
	EMO (22)	20 Hz	4 Hz

**Table 3 sensors-20-06535-t003:** Thresholds for the discretization of Russell’s *arousal-valence* model. If the self-reported value was equal or higher than a given threshold, it was considered as *high* (see the two columns); otherwise, it was *low*.

Dataset	High Arousal	High Valence
ASCERTAIN	≥3	≥0
AMIGOS	≥5	≥5
DECAF	≥2	≥0
WESAD	–	–

**Table 4 sensors-20-06535-t004:** Representation of classes in the datasets. For AMIGOS, DECAF, and ASCERTAIN, discretization of self-reported levels of arousal and values was done according to rules and thresholds in Table 3. For WESAD, the classes were assigned based on stimuli.

Dataset	LALV	LAHV	HALV	HAHV	Total
AMIGOS	80 (13%)	155 (25%)	194 (31%)	200 (32%)	629
DECAF	148 (6%)	714 (31%)	574 (25%)	843 (37%)	2279
ASCERTAIN	73 (4%)	221 (11%)	665 (34%)	982 (51%)	1941
		Amusement	Stress	Baseline	
WESAD	–	186 (17%)	332 (30%)	587 (53%)	1105

**Table 5 sensors-20-06535-t005:** Results for AMIGOS averaged over 5 iterations and 5 cross-validation folds ordered by F1-score.

Architecture	Accuracy	±	F1-Score	±	ROC AUC	±
Random guess	0.25	–	0.24	–	–	–
Resnet	0.31	0.04	0.23	0.04	0.55	0.02
FCN	0.31	0.04	0.22	0.04	0.55	0.03
Stresnet	0.27	0.04	0.22	0.04	0.52	0.03
Encoder	0.30	0.03	0.20	0.04	0.52	0.03
Inception	0.29	0.05	0.20	0.05	0.54	0.02
MLP-LSTM	0.29	0.02	0.16	0.02	0.50	0.03
Time-CNN	0.29	0.06	0.14	0.05	0.51	0.01
Majority class	0.35	–	0.13	–	–	–
MLP	0.31	0.03	0.13	0.03	0.50	0.01
MCDCNN	0.26	0.07	0.12	0.04	0.50	0.00
CNN-LSTM	0.13	0.00	0.06	0.00	–	–

**Table 6 sensors-20-06535-t006:** Results for DECAF averaged over 5 iterations and 5 cross-validation folds ordered by F1-score.

Architecture	Accuracy	±	F1-Score	±	ROC AUC	±
FCN	0.36	0.02	0.26	0.02	0.53	0.01
Stresnet	0.35	0.02	0.25	0.02	0.55	0.02
Encoder	0.37	0.02	0.25	0.02	0.54	0.02
Inception	0.35	0.02	0.25	0.02	0.53	0.02
Resnet	0.35	0.01	0.25	0.02	0.54	0.01
MLP-LSTM	0.37	0.02	0.24	0.03	0.55	0.01
Random guess	0.25	–	0.23	–	–	–
CNN-LSTM	0.37	0.01	0.23	0.02	0.54	0.01
MCDCNN	0.31	0.09	0.15	0.06	0.52	0.02
Majority class	0.38	–	0.14	–	–	–
MLP	0.34	0.04	0.13	0.01	0.50	0.00
Time-CNN	0.29	0.09	0.13	0.04	0.50	0.01

**Table 7 sensors-20-06535-t007:** Results for ASCERTAIN averaged over 5 iterations and 5 cross-validation folds ordered by F1-score.

Architecture	Accuracy	±	F1-Score	±	ROC AUC	±
Inception	0.47	0.02	0.24	0.01	0.51	0.02
Resnet	0.46	0.03	0.24	0.02	0.52	0.01
FCN	0.48	0.01	0.22	0.01	0.52	0.01
Stresnet	0.47	0.02	0.22	0.02	0.52	0.02
Encoder	0.50	0.01	0.22	0.02	0.52	0.02
Random guess	0.25	–	0.21	–	–	–
MLP-LSTM	0.50	0.01	0.20	0.02	0.52	0.01
Majority class	0.50	–	0.17	–	–	–
Time-CNN	0.42	0.11	0.16	0.03	0.50	0.01
MLP	0.46	0.06	0.16	0.01	0.50	0.00
MCDCNN	0.42	0.13	0.16	0.03	0.50	0.01
CNN-LSTM	0.04	0.00	0.02	0.00	–	–

**Table 8 sensors-20-06535-t008:** Results for WESAD averaged over 5 iterations and 5 cross-validation folds ordered by F1-score.

Architecture	Accuracy	±	F1-Score	±	ROC AUC	±
FCN	0.79	0.05	0.73	0.07	0.91	0.02
Resnet	0.74	0.07	0.69	0.07	0.89	0.04
Time-CNN	0.75	0.03	0.66	0.05	0.86	0.02
MCDCNN	0.74	0.04	0.62	0.05	0.84	0.03
Stresnet	0.69	0.11	0.62	0.10	0.82	0.05
MLP-LSTM	0.73	0.01	0.61	0.03	0.82	0.01
Inception	0.71	0.06	0.58	0.07	0.81	0.07
Encoder	0.71	0.03	0.57	0.05	0.83	0.02
MLP	0.72	0.01	0.57	0.01	0.78	0.02
CNN-LSTM	0.70	0.02	0.56	0.03	0.79	0.02
Random guess	0.33	–	0.32	–	–	–
Majority class	0.53	–	0.23	–	–	–

**Table 9 sensors-20-06535-t009:** F1-score, precision, recall, and support (number of samples) of models with the highest F1-score averaged over all iterations and folds for each class and dataset, separately. F1-scores from Table 5, Table 6, Table 7 and Table 8 are the arithmetic means of F1-scores calculated for each class in a given dataset. Please note that the F1-score in this table is an arithmetic mean of the harmonic means of precisions and recalls, so, e.g., for the baseline in WESAD, the F1-score is not simply a harmonic mean of precision and recall. Also support is averaged over all folds - multiplying it by 5 provides the values from Table 4.

Dataset	Class	F1	Precision	Recall	Support
WESAD	Baseline	0.80	0.87	0.79	117.4
WESAD	Stress	0.92	0.92	0.93	66.4
WESAD	Amusement	0.48	0.55	0.54	37.2
DECAF	LALV	0.00	0.00	0.00	29.6
DECAF	LAHV	0.33	0.35	0.34	142.8
DECAF	HALV	0.24	0.33	0.20	114.8
DECAF	HAHV	0.45	0.40	0.56	168.6
ASCERTAIN	LALV	0.01	0.02	0.00	14.6
ASCERTAIN	LAHV	0.03	0.08	0.02	44.2
ASCERTAIN	HALV	0.32	0.39	0.30	133.0
ASCERTAIN	HAHV	0.60	0.52	0.74	196.4
Amigos	LALV	0.01	0.01	0.02	16.0
Amigos	LAHV	0.26	0.27	0.31	31.0
Amigos	HALV	0.30	0.32	0.37	38.8
Amigos	HAHV	0.33	0.35	0.39	40.0

**Table 10 sensors-20-06535-t010:** Ranks and average rank are presented for each architecture. Although Friedman’s rank test yielded p=0.003, the post-hoc Wilcoxon-Holm method did not present any statistically significant differences between any given pair of architectures.

	AMIGOS	ASCERTAIN	DECAF	WESAD	Average
FCN	2	3	1	1	1.75
Resnet	1	2	5	2	2.5
Stresnet	3	4	2	5	3.5
Inception	5	1	4	7	4.25
Encoder	4	5	3	8	5
MLP-LSTM	6	6	6	6	6
Time-CNN	7	7	10	3	6.75
MCDCNN	9	9	8	4	7.5
MLP	8	8	9	9	8.5
CNN-LSTM	10	10	7	10	9.25

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
