# Peer review of "Can We Ditch Feature Engineering? End-to-End Deep Learning for Affect Recognition from Physiological Sensor Data"

_sensors, 2020, doi:10.3390/s20226535_

Round 1

Reviewer 1 Report

The paper consists of 21 pages. The length of paper is adequate for journal publication. Authors in section Introduction describe emotions and emotional states from historical point. This step is unnecessary, the section Introduction is for reader very long. Similarly problem is also in section Related Work. In this section authors describes various sensors and it application way of measuring. In this section missing the various ideas from others authors as we can the wristbands use for measuring and future classification the physiological data (from these wristbands). The presented and discussed methods in section Related Work are right and adequate.

For authors I have these questions:

  1. How authors synchronize measurement data from various sensors? Please add this passage in text to your paper.
  2. Were devices (wristbands) of the same type used? The various types they may have different in accuracy. How authors solved this problem?
  3. In section discussion the authors they state: ”…One reason for such behavior may be the fact that physiological responses induced by the emotional videos used in the emotion datasets is generally smaller than responses invoked by the Trier Social Stress Test used in WESAD. The difficulty of recognizing subtle affective stimuli using physiological sensors has also been recognized in the related work [12]…”  As we know, affect is very short (but expressive) demonstration of emotional state. In this paper missing time duration of video clips. In this moment is not possible to determine how long was evoked the specifically emotional state.

Reviewer 2 Report

The article presents a comparison of ten neural network architectures for emotion detection directly from the preprocessed signals without forma extraction of customized features. The task is emotion detection from signals collected with wearables. For evaluation are used four publicly available datasets. The hyper parameters of the proposed neural network architectures are formally optimized using Bayesian optimization. As evaluation parameters are selected accuracy, F1-score, ROC AUC. The evaluated architectures are ranked, and results discussed in detail. The paper is well written and easy to read and comprehend. Notes:

= Will be nice to have mentioned the state-of-the-art published results for the four datasets using the classic approach: formal feature extraction and then neural network-based classifier. Just for comparison purposes, it is OK if these results are better than the proposed approach. It will be very nice if authors can provide some results from using the traditional approach with hand-crafted features on the same datasets. It will just require looking for them in the corresponding papers. It is completely OK if these results are with better accuracy than with the proposed approach - it is understandable that it is a novel and may not provide same accuracy as already polished algorithms, but I believe it is prospective and applicable beyond emotion recognition.

= In references section, [2] has misspelled name of the first author, [15], [16], and maybe [33] have missing capitalizations in the paper/article title.

Reviewer 3 Report

To investigate the possibility of using multimodal deep learning for affect recognition, the authors tested 10 end-to-end architectures, in addition, a lot of experiments have been conducted in four open datasets. But there are still some aspects that need to be further improved: 1) The abstract says that in order to be able to make a fair comparison between different depth architectures, Bayesian optimization is used to adjust hyperparameters. Can the relationship between them be further explained? Please explain the benefit of Bayes. 2) Two (high/low flow level) or three flow states (boredom, stress, flow) in Line 169. Refers to the specific what are them? Please give a more detailed explanation. 3) What is the multimodal deep learning model in Figure 1, please give an explanation. 4) Is the training data obtained through investigation and research? How much data is there? Will there be overfitting caused by small number of training data? 5) The index of F1-SCORE has been used for many times in this paper, so it should be introduced first. 6) In Section 3.3, Equations 2-7 listed separately for each network are almost the same except the parameters. Please explain how these parameters were selected. 7) This paper introduces 10 network architectures, and uses 4 data sets for experiments, but does not specify how to use these network structures for Affect Recognition. 8) Overall, the literature review should be improved as it neglects some efficient and up-to-date methods. Without discussing the previous works, the novelty of the paper cannot be assessed properly. It is suggested that the following papers should be cited: [1] DOI: 10.1007/s12652-020-02572-0  [2] DOI: 10.1109/ACCESS.2020.2972338 [3] DOI: 10.1109/ACCESS.2019.2924944 [4] DOI: 10.3390/s20041188

Round 2

Reviewer 3 Report

No more comments.